# Epidemiological, virological and serological investigation of a SARS-CoV-2 outbreak (Alpha variant) in a primary school: A prospective longitudinal study

Elsa Lorthe[1]ᴼ*, Mathilde Bellon[2,3]ᴼ, Grégoire Michielin[4], Julie Berthelot[1], María-Eugenia Zaballa[1], Francesco Pennacchio[1], Meriem Bekliz[2], Florian Laubscher[5], Fatemeh Arefi[4], Javier Perez-Saez[1,6], Andrew S. Azman[1,6,7], Arnaud G. L'Huillier[5,8], Klara M. Posfay-Barbe[8], Laurent Kaiser[3,9], Idris Guessous[10,11], Sebastian J. Maerkl[4], Isabella Eckerle[2,3,5,9]‡, Silvia Stringhini[1,11,12]‡, on behalf of the SEROCoV-Schools Study Group¶

1 Unit of Population Epidemiology, Division of Primary Care Medicine, Geneva University Hospitals, Geneva, Switzerland, 2 Department of Microbiology and Molecular Medicine, Faculty of Medicine, University of Geneva, Geneva, Switzerland, 3 Center for Emerging Viral Diseases, Geneva University Hospitals and University of Geneva, Geneva, Switzerland, 4 Institute of Bioengineering, School of Engineering, École Polytechnique Fédérale de Lausanne, Lausanne, Switzerland, 5 Laboratory of Virology, Department of Diagnostics, Geneva University Hospitals, Geneva, Switzerland, 6 Department of Epidemiology, Johns Hopkins Bloomberg School of Public Health, Baltimore, Maryland, United States of America, 7 Institute of Global Health, Faculty of Medicine, University of Geneva, Geneva, Switzerland, 8 Department of Pediatrics, Gynecology & Obstetrics, Pediatric Infectious Disease Unit, Geneva University Hospitals and Faculty of Medicine, Geneva, Switzerland, 9 Division of Infectious Diseases, Department of Medicine, Geneva University Hospitals, Geneva, Switzerland, 10 Division of Primary Care, Geneva University Hospitals, Geneva, Switzerland, 11 Department of Health and Community Medicine, Faculty of Medicine, University of Geneva, Geneva, Switzerland, 12 University Center for General Medicine and Public Health, University of Lausanne, Lausanne, Switzerland

ᴼ These authors contributed equally to this work.
‡ IE and SS also contributed equally to this work and consider as senior authors.
¶ Membership of the SEROCoV-Schools Study Group is provided in the Acknowledgments.
* elsa.lorthe@gmail.com

**Data Availability Statement:** Despite all the precautions taken, our data contain potentially identifying information due to the nature of this

## Abstract

### Objectives

To report a prospective epidemiological, virological and serological investigation of a SARS-CoV-2 outbreak in a primary school.

### Methods

As part of a longitudinal, prospective, school-based surveillance study, this investigation involved repeated testing of 73 pupils, 9 teachers, 13 non-teaching staff and 26 household members of participants who tested positive, with rapid antigen tests and/or RT-PCR (Day 0–2 and Day 5–7), serologies on dried capillary blood samples (Day 0–2 and Day 30), contact tracing interviews and SARS-CoV-2 whole genome sequencing.

### Results

We identified 20 children (aged 4 to 6 years from 4 school classes), 2 teachers and a total of 4 household members who were infected by the Alpha variant during this outbreak. Infection

study and the fact that we are reporting information on relatively small groups of children from a specific geographic location. Therefore, the study steering committee members decided to make these data accessible to researchers who meet the criteria for access to confidential data upon reasonable request for data sharing to the Unit of Population Epidemiology (uep@hcuge.ch). All requests will be evaluated by the Data Access Committee and approved on the basis of their scientific quality.

**Funding:** Funding: The SEROCoV-Schools study was supported by the Federal Office of Public Health, the Private Foundation of the Geneva University Hospitals, the Fondation des Grangettes, the Center for Emerging Viral Diseases, and a SNF NRP (National Research Program) 78 Covid-19 Grant 198412 (to S.J.M., I.E.). The funders had no role in study design, data collection and analysis, decision to publish, or preparation of the manuscript.

**Competing interests:** The authors have declared that no competing interests exist.

attack rates were between 11.8 and 62.0% among pupils from the 4 school classes, 22.2% among teachers and 0% among non-teaching staff. Secondary attack rate among household members was 15.4%. Symptoms were reported by 63% of infected children, 100% of teachers and 50% of household members. All analysed sequences but one showed 100% identity. Serological tests detected 8 seroconversions unidentified by SARS-CoV-2 virological tests.

## Conclusions

This study confirmed child-to-child and child-to-adult SARS-CoV-2 transmission and introduction into households. Effective measures to limit transmission in schools have the potential to reduce the overall community circulation.

## Introduction

Children play an important role in the transmission of many respiratory viral diseases, including beta-coronaviruses and influenza virus, both within schools [1] and within households [2, 3]. This has led most countries worldwide to implement school closures as one component of severe acute respiratory syndrome coronavirus 2 (SARS-CoV-2) transmission mitigation policies from the very beginning of the pandemic [4].

Young children commonly have fewer and milder symptoms of SARS-CoV-2 infection than adults, with a high proportion of asymptomatic infections, and are less likely to experience severe coronavirus disease (COVID-19) [5]. However, epidemiological and biological data suggest that, when infected, children transmit as much as adults, as children achieve viral loads comparable, or only minimally lower, to those of adults at the time of diagnosis [6–10].

At school, young children have many prolonged close contacts with peers and adults indoors [11], usually do not wear masks, and in many countries they are not systematically tested when symptomatic, including in Switzerland [12]. These circumstances make children and schools a potential strong contributor of the overall community SARS-CoV-2 transmission [13]. Published studies reported school outbreaks in the United States [14–16], Australia [17], England [18], Ireland [19], Norway [20] or Italy [21], and/or seroprevalence estimates in schools in Switzerland [22, 23], Canada [24] or Germany [25, 26]. Overall, they suggested a relatively low incidence of COVID-19 in children or low prevalence of anti-SARS-CoV-2 antibodies, conveying the message that schools are not a high risk setting for transmission of COVID-19 [27, 28]. However, most studies were retrospective, had limited participation rates, were performed before the emergence of variants of concern, and did not systematically test both symptomatic and asymptomatic children and adults. Therefore, the extent to which young children are infected and transmit SARS-CoV-2 in school settings remains controversial [5, 29, 30], in particular with variants of concern. Evidence on transmission direction (adult-to-child, child-to-child, child-to-adult) is also sparse.

We aim to report a prospective epidemiological, virological and serological investigation of a SARS-CoV-2 outbreak in a primary school in Geneva, Switzerland, in April-May 2021.

## Materials and methods

### Study design

This outbreak investigation is part of a longitudinal, prospective, surveillance study (SERO-CoV-Schools), which aims to describe the transmission dynamics of SARS-CoV-2 infection

within primary schools and early childhood education centres in a sample of five institutions in Geneva, and the risk of introduction of SARS-CoV-2 into the children's households. The study started in March 2021. Participants (pupils and their teachers) had a baseline assessment which included SARS-CoV-2 serology from a capillary blood test, an antigen rapid diagnostic test (RDT) from an oropharyngeal swab sample, and the completion of an online questionnaire. Then, a surveillance phase started, with weekly questionnaires and self-declarations (anytime outside of the weekly questionnaires) allowing participants to report COVID-19-like symptoms, contact with a positive case or the diagnosis of a SARS-CoV-2 infection. An outbreak investigation was triggered when a positive case was diagnosed from a positive real-time reverse transcription polymerase chain reaction (RT-PCR), which was the case on April 26, 2021. The date of diagnosis of the first positive case in a school class was referred to as Day 0 for that school class. The investigation involved repeated virological and serological testing of the participants: RDT and/or RT-PCR were performed at Day 0–2 and Day 5–7, serologies were performed at Day 0–2 and Day 30.

## Study population

In the investigated school, all 9 school classes with children aged 3 to 6–7 years old participated in the SEROCoV-Schools study. The present study population consisted of children and school staff in school classes with a positive case as well as household members of the confirmed cases. School staff included teachers (including assistants) and non-teaching staff (administrative, cleaners, catering). Two children in Class 2 were absent from school for the 2 weeks preceding this outbreak, therefore they were considered as non-exposed and not included in the analysis.

## Context and public health measures

During the period of this outbreak investigation, non-pharmaceutical public health measures in Geneva were gradually relaxed, with restaurants and bars opening their outdoor spaces and entertainment venues opening their indoor spaces.

Mitigation of the COVID-19 pandemic in Switzerland included school closures during the first wave from March to May 2020. Thereafter, priority was given to keeping schools open with several types of preventive measures in place which varied widely across institutions. In the investigated school, measures in place during the outbreak included checking children's temperature every morning, sending children home if they had fever or sickness beyond very mild symptoms, restricted access for parents and requiring all adults (but not young children) to wear facemasks.

## Epidemiological investigation

**Case definition.** Cases were defined according to laboratory results: confirmed SARS-CoV-2-infected cases were those with positive SARS-CoV-2 RDT and/or RT-PCR results, and/or a seroconversion between Day 0–2 and Day 30 (from seronegative to seropositive according to the test-specific cut-off, unrelated to vaccination). Confirmed cases could be further classified as symptomatic or asymptomatic.

**Outcomes.** Only descriptive analyses are presented here. Infection attack rates (IAR) were defined as the number of children and school staff with a positive RDT and/or RT-PCR detected within a 21-day period of the index case identification, and/or a seroconversion, divided by the total number of children and school staff participating in the study. Household secondary attack rates (SAR) were defined as the number of non-index household members who tested positive by RDT and/or RT-PCR within a 21-day period of the first infection

detected in each household or who seroconverted, divided by the total number of non-index members participating in the study. Adults were defined as individuals ≥18 years, whereas children were defined as individuals <18 years of age. Plausible directions of transmission were determined, when possible, on the basis of symptom onset and testing dates.

**Contact tracing.** Positive cases (or their parents in the case of children) were interviewed using a structured questionnaire investigating symptoms, contact with a positive person, school attendance, extracurricular activities, play dates/birthday parties, family/friend gatherings and best friends, in the 14 days prior to diagnosis.

## Laboratory investigation

**Virological testing.** SARS-CoV-2 testing was performed on oropharyngeal swabs taken by nurses or medical doctors of the study team. This swabbing method is less invasive than nasopharyngeal swabs and more acceptable by young children. Children who were not attending school on the testing days were tested by their paediatrician or at a testing centre and communicated the results to our team. Depending on the visit, we performed RT-PCR or RDT tests. We used the Panbio COVID-19 Ag rapid test (Abbott) which has been validated in adults for use with oropharyngeal instead of nasopharyngeal swabs [31]. All oropharyngeal swab samples used for the RDT were tested a second time from the same swab by RT-PCR to confirm the result (in-house SARS-CoV-2 RT-qPCR or Cobas® SARS-CoV-2 Test, Cobas 6800, Roche, Switzerland). SARS-CoV-2 whole genome sequencing was performed for positive samples at the Health 2030 Genome Center (Geneva) using the Illumina COVIDSeq library preparation reagents following the protocol provided by the supplier. All results were communicated to the participants as soon as they became available.

**Serological testing.** We collected capillary blood on a Neoteryx Mitra® collection device at Day 0–2 and Day 30, and tested for anti-Spike-SARS-CoV-2 IgG on a microfluidic nanoimmunoassay as described previously [32]. This sampling method is minimally invasive, easily implementable in schools and more acceptable than traditional venipuncture by children. Serologies were communicated to the families approximately 2 months after the visit.

## Ethics

This study was approved by the ethics committee of the Canton of Geneva (Project ID 2020–02957). All parents and teachers were informed about the study (orally and through a flyer and a detailed information note) and gave written consent. We also provided a short letter written in a child-friendly style, and a schema illustrating concepts of infection and antibodies, which parents could use to inform their child. Before each test, a team of trained nurses discussed with each child, explained the tests to be taken and answered any questions. Children gave verbal assent to participate.

## Results

### Outbreak description, epidemiological and serological investigation

The first COVID-19 case (index case: Teacher 1 [T1], Class 1) was diagnosed by RT-PCR performed in a laboratory on April 26, 2021 (Fig 1), 3 days after the onset of symptoms (April 23). This participant reported the result to our team, which triggered the investigation. The second teacher in Class 1 (T2) had symptoms onset on April 24 with an initially negative RT-PCR on April 26 followed by a positive RT-PCR on April 28 (Fig 2). Both attended school until April 23 included.

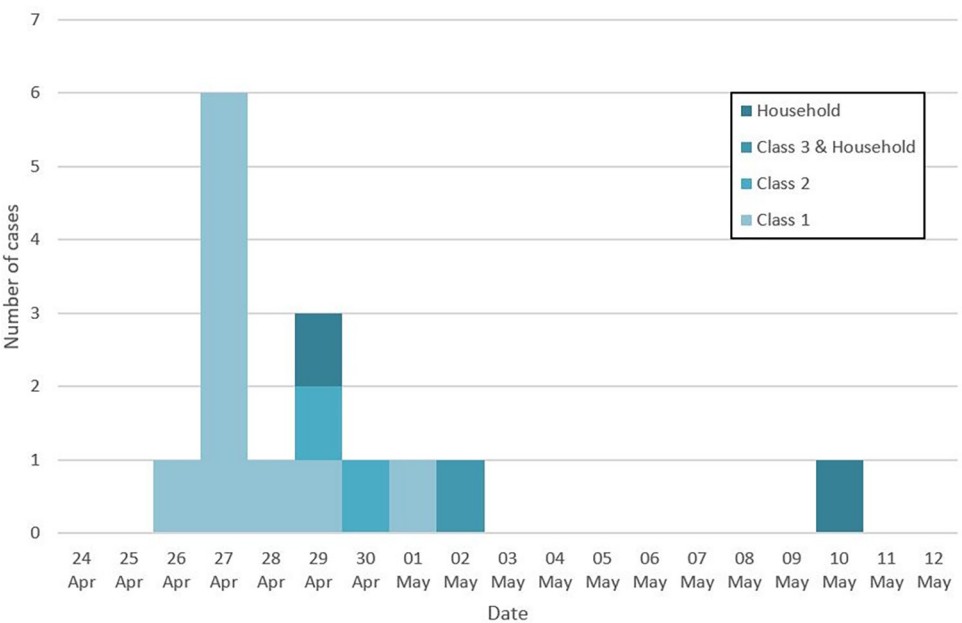

**Fig 1. Epidemic curve of cases confirmed by positive RDT or RT-PCR (n = 15), school outbreak, Geneva, Switzerland, 2021.** Apr: April.

All 21 children from Class 1, aged 4 years old, were tested on April 27, 2021. They all attended school the week before. The IAR was 62% (13/21): 1 child had a positive RDT confirmed by a RT-PCR (Kid 1 [K1]); 5 children had a negative RDT, but subsequent RT-PCR testing on the same swab samples came back positive (K2, K3, K8, K11, K12); 2 children absent from school because symptomatic were tested by healthcare providers outside the study setting, both had a positive RT-PCR (K4, K5); 1 child had a positive RT-PCR at the second visit at Day 7 (K10); and 4 children seroconverted between the first and the last visit at D30 (K13, K14, K15, K16) despite negative swab virological tests at Day 1 and Day 7 (Table 1, Figs 1–3).

Children in another school class from the same grade (Class 2) were identified as having close contacts with children of Class 1 during breaks and specific activities. Class 2 was therefore tested on April 29, 2021. Of the 19 children aged 4 years old, 17 were tested by RT-PCR, of whom 2 (K6, K7) tested positive (IAR 11.8%), and 2 did not participate in the outbreak investigation (they both had only a negative RT-PCR performed externally on May 6). No additional case was identified at subsequent visits. We also repeatedly tested 4 teachers and 2 non-teaching staff in contact with Classes 1 and 2, none of whom was diagnosed with a SARS-CoV-2 infection (Table 1).

Of note, all children from these two school classes were placed in quarantine by local health authorities for 10 days from April 30, 2021.

The sibling of a positive case from Class 1, attending Class 3 in the same school, tested positive on May 2 (K9), triggering the testing protocol in Class 3. All 17 6-year old tested in Class 3 had a negative RT-PCR on May 4, and 16/16 had a negative RDT on May 11 (subsequently confirmed with a RT-PCR). Serological tests at Day 30 revealed 2 additional seroconversions (K17, K20, of whom one was not tested at Day 2 of the outbreak but seroconverted since baseline at the end of March 2021) (IAR 16.7%). In the other school class from the same grade, Class 4, tested because of close and regular contacts with Class 3, 15/15 had a negative RT-PCR on May 4, 13/13 had a negative RDT on May 11, and 2 additional cases of seroconversion

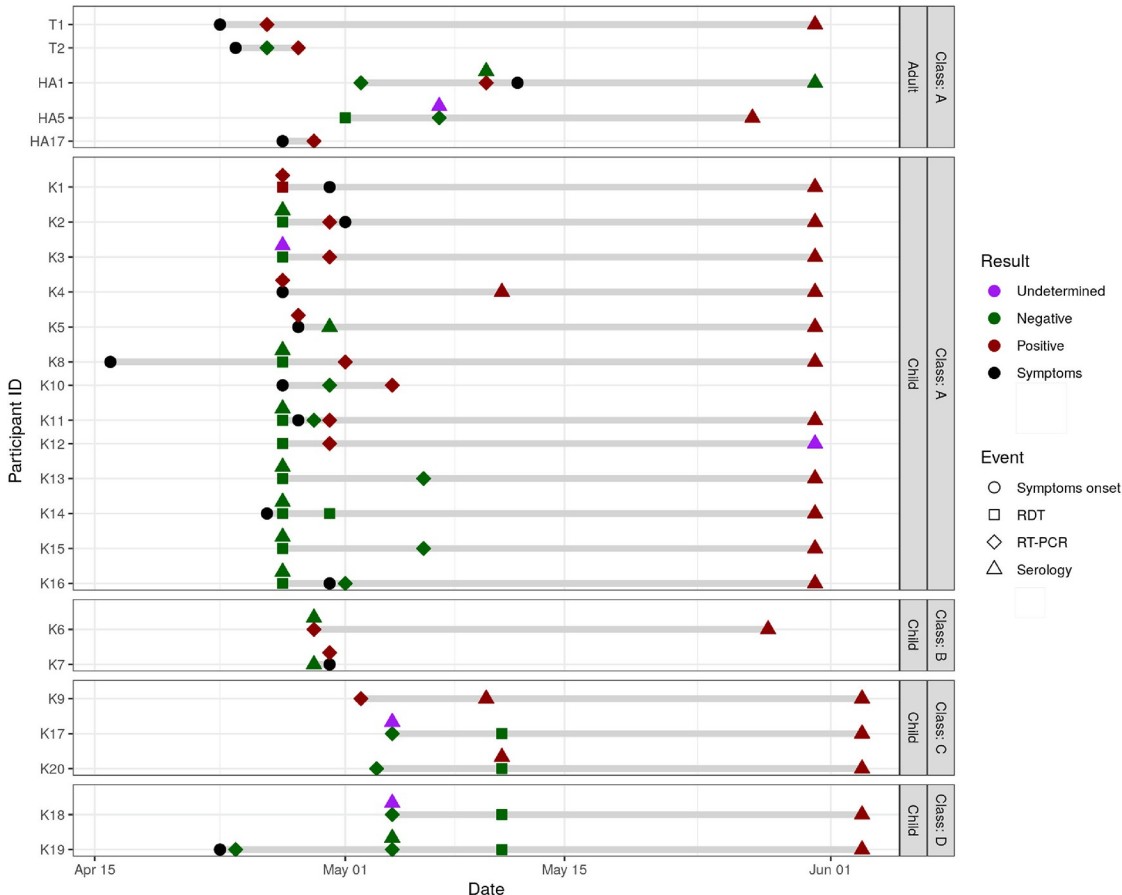

**Fig 2. Timeline of symptoms onset, diagnosis and virological analyses among cases with a confirmed SARS-CoV-2 infection (positive RDT, RT-PCR or seroconversion).** Apr: April, HAx: Adult member of household, HCx: Child member of household, Jun: June, Kx: Kid x, Tx: Teacher x, RDT: antigen rapid diagnostic test, RT-PCR: real-time reverse transcription polymerase chain reaction.

(K18, K19) were identified (IAR 13.3%); 2 children in the school class did not take part in the study (Table 1, Fig 2). Among 3 teachers and 11 non-teaching staff, none tested positive.

Transmission occurred in 3 out of 10 investigated households of participants with a positive SARS-CoV-2 test, leading to a secondary attack rate among household members of 4/26 (15.4%). Three members of the same household refused to participate in this investigation, two of whom were vaccinated. Secondary attack rates were 1/2 (50.0%) with an adult index case, and 3/24 (12.5%) with a child index case. A teacher spent a few days at his/her parents' place while having symptoms, his/her parent then developed symptoms and tested positive (adult household member [HA17]). The parent (HA5) of a positive child from Class 1 tested negative twice (RDT on May 1, RT-PCR on May 7), though he/she seroconverted (Figs 2 and 3). He/she reported no contacts outside his/her household, strictly followed all recommendations, and was not vaccinated between the two blood draws. Finally, in the family of two children from Classes 1 and 3 who tested positive one after the other, a parent (HA1) who initially tested negative by RT-PCR, tested positive after quarantine with his/her children.

Among the 15 cases with a positive SARS-CoV-2 test, 3/3 (100%) adults and 9/12 (69%) children reported symptoms either before or after the positive test, and 3 children were asymptomatic. Among the 8 children who seroconverted without a positive RDT or RT-PCR test, 3

**Table 1. Overview of tests results, symptoms and cases among the 4 investigated classes, teachers and non-teaching staff.**

| | Visit 1 (Day 0–2) | | Visit 2 (Day 5–7)[a] | | Seroconversion (D30)[b] | Symptoms[c] | Total cases[d] |
|---|---|---|---|---|---|---|---|
| | RDT + n positive/ n tested | RT-PCR + n positive/n tested | RDT + n positive/ n tested | RT-PCR + n positive/n tested | n seroconversions/N tested | n/n confirmed cases | n confirmed cases/ n tested |
| Class A | 1/17 | 8[e]/21 | 0/1 | 1/16 | 4/11 | 10/13 | 13/21 |
| Class B | -/- | 2/17 | -/- | 0/15 | 0/12 | 1/2 | 2/19 |
| Class C | 0/1 | 1/17 | 0/14 | 0/2 | 2/16[f] | 0/3 | 3/18 |
| Class D | -/- | 0/15 | 0/12 | 0/1 | 2/10 | 1/2[g] | 2/15 |
| Teachers | 0/4 | 2/9 | 0/2 | 0/3 | 0/7 | 2/2 | 2/9 |
| Non-teaching staff | 0/2 | 0/10 | 0/4 | 0/2 | 0/10 | -/- | 0/13 |

[a] Tests were not repeated in participants with a SARS-CoV-2 infection diagnosed at Visit 1

[b] Among participants with negative swab tests both at Visit 1 and Visit 2, and negative serology at baseline and/or at Visit 1. Adults who were vaccinated and developed antibodies between visit 1 and visit 3 were not considered as related to the outbreak.

[c] Among participants with a SARS-CoV-2 positive test and/or a seroconversion

[d] A case was defined as a participant with a positive RDT and/or a positive RT-PCR and/or a seroconversion.

[e] Including confirmation of one positive RDT by a subsequent RT-PCR performed on the same day

[f] Including 1 seroconversion between March 26, 2021 and May 11, 2021

[g] No data on symptoms for one child

were symptomatic, 4 were asymptomatic, and one did not provide symptom information. Overall, 12/19 (63%) infected children reported symptoms. No severe form of COVID-19 (requiring hospitalization) was reported, and all cases recovered well.

## Contact tracing analysis

The two infected teachers live alone and reported no contact with a positive or symptomatic case in their private life or activities, during the 14 days prior to infection. Their presumed source of infection was school, as several children from Class 1 were coughing during the week of April 19. The two infected parents were likely infected by their children, as they reported no contact outside the household.

Three social activities outside the school setting occurred in the two weeks before the triggering case was diagnosed: the first one (April 18, outdoors) brought together 6 children from Class 1 of whom 5 subsequently tested positive and 3 kids from another school (not included in the study); the second one (April 21, both indoors and outdoors) gathered 4 children from Class 1 and Class 2 of whom 3 tested positive (the other one had antibodies at baseline and at Visit 1); the last one (April 24, outdoors) was attended by 2 kids from Class 1 and 2 kids from another school (not included in the study), 1 tested positive at the 2nd visit.

## Virological investigation

We conducted whole genome sequencing on samples from the 15 participants who tested positive for SARS-CoV-2. Coverage of three isolate genomes (HA1, K9 and K10) were insufficient for any comparisons due to low viral load in the specimen (32%, 20% and 16% coverage only, respectively). The other sequenced genomes belonged to the Alpha variant. The virus sequence of K12 differed from the others in one region covered. Here, in position 15824–15827 a deletion and one addition that restores the reading frame was observed, resulting in a total of 4 nucleotides difference compared to the other sequences. All 9 sequences with >99% coverage shared 100% identity between genomes in comparison to the reference sequence (Fig 4). Virus

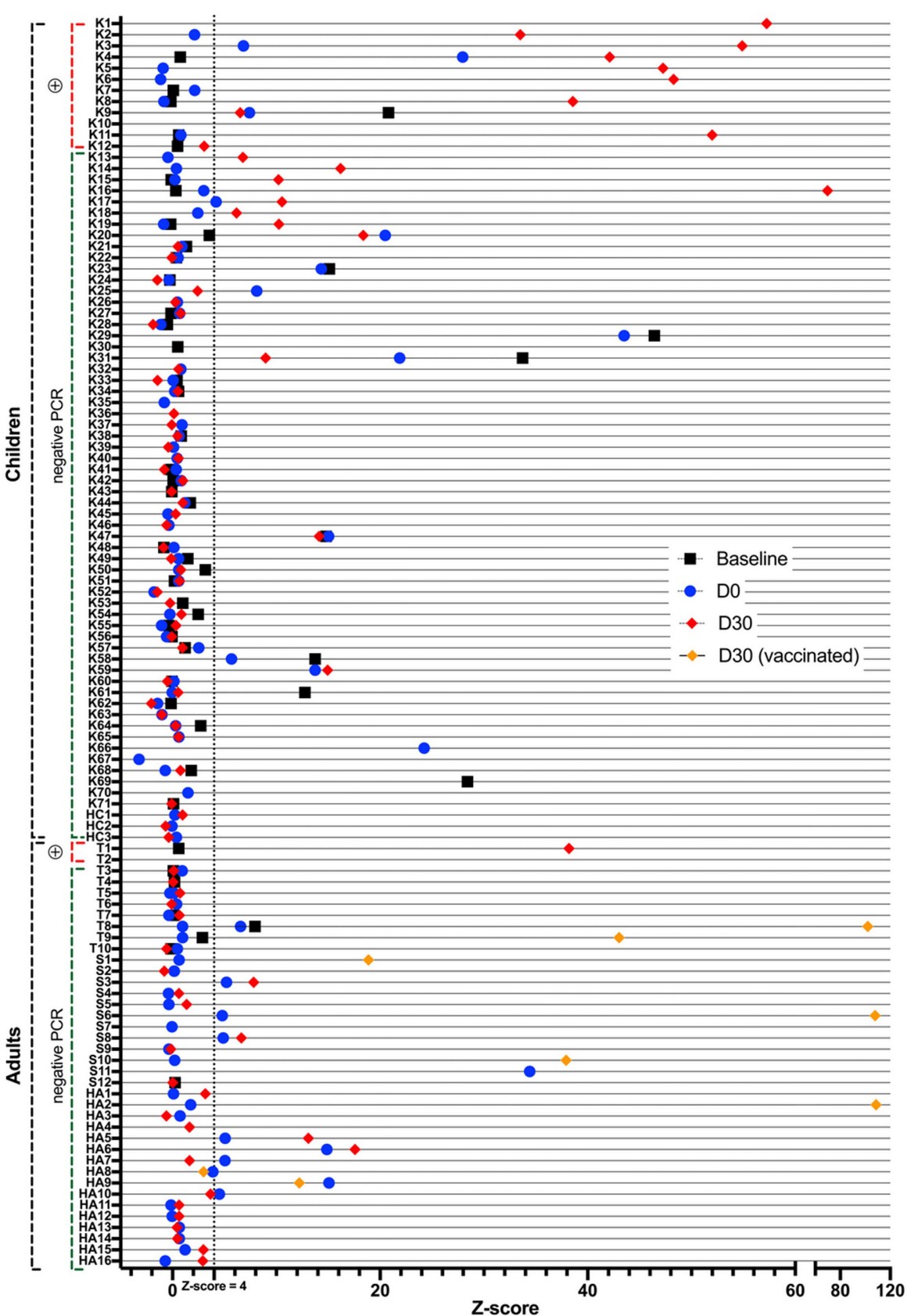

**Fig 3. Results of serological tests at baseline, at D0-2 and D30 for all pupils, teachers, non-teaching staff and household members included in the outbreak investigation.** This figure displays the results of the serological tests performed at baseline (black square), i.e. at the beginning of the study in March 2021, and during the outbreak investigation at Day 0–2 (blue circle) and Day 30 (red diamond). Adults who were vaccinated during the outbreak investigation are indicated by a yellow diamond. Household members who had no serological test are not represented. D: Day, HAx: Adult member of household, HCx: Child member of household, Kx: Kid x, Tx: Teacher x, PCR: real-time reverse transcription polymerase chain reaction.

specimens that could only be partially sequenced retrieved the same sequence without additional mutations in the regions covered.

## Discussion

This is one of the first investigation of a SARS-CoV-2 outbreak caused by the Alpha variant in a primary school. It involved 20 children from 4 school classes, 2 teachers and 4 household members (of whom one triggered the investigation in another school class). The primary case could not be formally identified, but it is likely that this outbreak started during the week of April 19 and was only identified on April 26 when a teacher tested positive and triggered the investigation. Of note, children in this age group were not routinely tested by the official testing recommendations in Switzerland when symptomatic [12]. This prospective school class-based study provides evidence of SARS-CoV-2 circulation among young children, school teachers, and introduction into households.

Since at least 9 positive cases of this outbreak shared viruses with identical sequence, we conclude that they are part of the same cluster. This could have been either by simultaneous infection through the same source, or transmission chains between affected individuals. One divergent virus sequence was found in one of the positive children, which could be either a *de novo* mutation occurring during the outbreak, or constitute independent introductions into the school, with one leading to this cluster. It cannot be ruled out that, by coincidence, several infection events with viruses sharing the same sequence were introduced from the community independently into the school. However, this hypothesis seems less likely given the number of cases involved, the epidemiological link and time frame, and the limited period of potentially other exposures before the quarantine decision. Overall community circulation at the cantonal level was also relatively low at the time (weekly incidence: about 200 cases/100,000 inhabitants). Between April 15 and May 15, the Alpha variant was causing 92% of new SARS-CoV-2 infections in Geneva [33].

**Fig 4. Virological analysis of positive cases by SARS-CoV-2 full genome sequencing.** Mutations in comparison to the reference sequence (NC_045512) are highlighted in orange. Green fields indicate no mutation; grey fields indicate insufficient genome coverage; and yellow fields indicate mixed viral population of the two nucleotides given. Numbers indicate nucleotide positions; asterisks (*) mark lineage-defining mutations for the Alpha variant. ID: Identifier, GISAID: Global Initiative on Sharing Avian Influenza Data, Kx: Kid x, Tx: Teacher x.

Other main insights are as follows. First, viral circulation of the Alpha variant in young children aged 4–6 years old was high, with a majority of unspecific and mild symptomatic infections, which might explain relatively high secondary attack rates [13]. The observed cluster probably started in one class and spread to two other classes, most likely through direct child-to-child contacts and interactions as per our epidemiological investigation, although there could be non-sampled child or adult intermediaries. We identified two seroconversions in a fourth class, but with no identification of SARS-CoV-2 virus, thereby limiting the conclusions on transmission linked to the other school classes. Similar patterns were found in a prospective study of three B.1.1.7 variant outbreaks which occurred in three childcare centers in Germany in January–February 2021: 11/12 groups had secondary cases, and secondary attack rates ranged from 7 to 64% among groups of children aged 1–6 years [34].

Second, child-to-parent transmission occurred in two different households, and child-to-teacher transmission is probable, which supports previous findings [18, 34–36]. Of note, vaccination was opened to all persons aged 45 years or older from April 12, and to all people in the 16–44 age group from May 19. Therefore, only a minority of adults (1 household member and 1 non-teaching staff) were vaccinated at the beginning of the outbreak. Child-to-adult transmission seems to depend on the duration of contacts, as no infection was identified among non-teaching staff who spent only limited periods of time (meals) with children. This is contrary to previous findings [37], and may be explained by the young age of our participants and their behaviour involving physical proximity with their teachers [38].

Third, social activities outside the school could have contributed to the spread of the infection, as previously reported [39, 40]. However, they could also reflect the bonding between children and a closer contact at school, thus facilitating transmission.

Fourth, we evidenced low sensitivity of RDTs with an oropharyngeal specimen for identifying both symptomatic and asymptomatic infected children, which confirms previous results showing that only children with high viral load are identified by such tests [23, 41]. We conclude that RDTs with an oropharyngeal specimen are not the most appropriate for surveillance and/or outbreak investigation purposes. The analysis of joint RT-PCR/serological data shows a substantial under-detection of infections in young children, even with RT-PCR testing, although the optimal time point for viral testing might have been missed in the fixed testing scheme of this study. This finding is consistent with prospective research based on comprehensive testing protocols which identified a higher incidence of COVID-19 among children and higher rates of transmission than previously reported, including in households [14, 35, 36]. Under-detection of acutely infected children may be an explanation for the discrepancy between official numbers of infected children [42] and studies on virus prevalence and seroprevalence in school settings [24, 28, 43–47].

As a take-home message, we suggest that routinely reported infections are likely to be just the tip of the iceberg, highlighting the importance of testing contacts, regardless of symptoms, to identify SARS-CoV-2 infections and stop transmission by rapidly isolating positive cases.

## Strengths and limitations

Few large SARS-CoV-2 outbreaks in young children in school settings have been documented so far [48, 49] even fewer involved an investigation of variants of concern. As part of an ongoing prospective study, this investigation started less than 24 hours after the first case was laboratory-confirmed and involved the use of three complementary approaches. We followed up and repeatedly tested all contacts within four school classes regardless of symptoms. Repeated serological tests proved useful to retrieve seroconversions following asymptomatic or undiagnosed infections. However, we relied on a limited number of cases, even though, by definition,

the sample size of a school class outbreak can only be as large as the number of pupils in each school class. Not all children and adults were tested, which could lead to underestimating IAR and SAR. We might have missed infections among adults who were vaccinated between Day 0–2 and Day 30, as we could not distinguish between antibodies due to vaccination and those due to infection. Also, we could not test the household members of cases only detected by sero-conversion at Day 30 with no positive RT-PCR/RDT, leading to a potential underestimation of secondary attack rates. The study was performed before the circulation of the more infectious Delta and Omicron variants; estimates are therefore likely to be underestimated [50–53].

## Conclusion

This prospective school-based study provides evidence of SARS-CoV-2 transmission among young children and school teachers and introduction into households. Epidemiological investigation confirmed child-to-child and child-to-adult direction of transmission of the infection. Children may be a significant source of extra-household infections and have the potential to play a role in community transmission, potentially even more so with more contagious variants. With most of the adult and adolescent population vaccinated, young children could act as disease reservoirs. Effective strategies are needed to limit transmission in school settings and vaccination of school staff and children should be encouraged.

## Acknowledgments

We are very grateful to the children and teachers participating in the SEROCoV-Schools study and their families. We would like to thank the school staff for their responsiveness and support during the outbreak investigation. The authors also would like to thank the members of the Unit of Population Epidemiology for their daily support in all the tasks required by the SERO-CoV-School study and the Virology Laboratory staff for performing RT-PCRs and for helping with the storage of the samples.

## SEROCoV-Schools Study Group:

Elsa Lorthe (lead author, elsa.lorthe@gmail.com, Unit of Population Epidemiology, Division of Primary Care Medicine, Geneva University Hospitals, Geneva, Switzerland), Julie Berthelot (Unit of Population Epidemiology, Division of Primary Care Medicine, Geneva University Hospitals, Geneva, Switzerland), Maria-Eugenia Zaballa (Unit of Population Epidemiology, Division of Primary Care Medicine, Geneva University Hospitals, Geneva, Switzerland), Hélène Baysson (Unit of Population Epidemiology, Division of Primary Care Medicine, Geneva University Hospitals, Geneva, Switzerland; Department of Health and Community Medicine, Faculty of Medicine, University of Geneva, Geneva, Switzerland), Andrea Jutta Loizeau (Unit of Population Epidemiology, Division of Primary Care Medicine, Geneva University Hospitals, Geneva, Switzerland), Ania Wisniak (Unit of Population Epidemiology, Division of Primary Care Medicine, Geneva University Hospitals, Geneva, Switzerland; Department of Health and Community Medicine, Faculty of Medicine, University of Geneva, Geneva, Switzerland), Stéphanie Testini (Unit of Population Epidemiology, Division of Primary Care Medicine, Geneva University Hospitals, Geneva, Switzerland), Khadija Samir (Unit of Population Epidemiology, Division of Primary Care Medicine, Geneva University Hospitals, Geneva, Switzerland), Natalie Francioli (Unit of Population Epidemiology, Division of Primary Care Medicine, Geneva University Hospitals, Geneva, Switzerland), Severine Harnal (Unit of Population Epidemiology, Division of Primary Care Medicine, Geneva University Hospitals, Geneva, Switzerland), Deborah Urrutia Rivas (Unit of Population Epidemiology, Division of Primary Care Medicine, Geneva University Hospitals, Geneva, Switzerland), Javier

Perez-Saez (Unit of Population Epidemiology, Division of Primary Care Medicine, Geneva University Hospitals, Geneva, Switzerland; Department of Epidemiology, Johns Hopkins Bloomberg School of Public Health, Baltimore, USA), Nick Pullen (Unit of Population Epidemiology, Division of Primary Care Medicine, Geneva University Hospitals, Geneva, Switzerland), Francesco Pennacchio (Unit of Population Epidemiology, Division of Primary Care Medicine, Geneva University Hospitals, Geneva, Switzerland), Julien Lamour (Unit of Population Epidemiology, Division of Primary Care Medicine, Geneva University Hospitals, Geneva, Switzerland), Richard Dubos (Unit of Population Epidemiology, Division of Primary Care Medicine, Geneva University Hospitals, Geneva, Switzerland), Gaëlle Bryand-Rumley (Unit of Population Epidemiology, Division of Primary Care Medicine, Geneva University Hospitals, Geneva, Switzerland), Claire Semaani (Unit of Population Epidemiology, Division of Primary Care Medicine, Geneva University Hospitals, Geneva, Switzerland), Viviane Richard (Unit of Population Epidemiology, Division of Primary Care Medicine, Geneva University Hospitals, Geneva, Switzerland), Roxane Dumont (Unit of Population Epidemiology, Division of Primary Care Medicine, Geneva University Hospitals, Geneva, Switzerland), Prune Collombet (Unit of Population Epidemiology, Division of Primary Care Medicine, Geneva University Hospitals, Geneva, Switzerland; Department of Health and Community Medicine, Faculty of Medicine, University of Geneva, Geneva, Switzerland), Natacha Noël (Unit of Population Epidemiology, Division of Primary Care Medicine, Geneva University Hospitals, Geneva, Switzerland), Patrick Bleich (Unit of Population Epidemiology, Division of Primary Care Medicine, Geneva University Hospitals, Geneva, Switzerland), Nacira El Merjani (Unit of Population Epidemiology, Division of Primary Care Medicine, Geneva University Hospitals, Geneva, Switzerland), Caroline Pugin (Unit of Population Epidemiology, Division of Primary Care Medicine, Geneva University Hospitals, Geneva, Switzerland), Jessica Rizzo (Unit of Population Epidemiology, Division of Primary Care Medicine, Geneva University Hospitals, Geneva, Switzerland), Marion Frangville (Unit of Population Epidemiology, Division of Primary Care Medicine, Geneva University Hospitals, Geneva, Switzerland), Antoine Bal (Unit of Population Epidemiology, Division of Primary Care Medicine, Geneva University Hospitals, Geneva, Switzerland), Fanny-Blanche Lombard (Unit of Population Epidemiology, Division of Primary Care Medicine, Geneva University Hospitals, Geneva, Switzerland), Zo Francia Randrianandrasana (Unit of Population Epidemiology, Division of Primary Care Medicine, Geneva University Hospitals, Geneva, Switzerland), Oumar Aly Ba (Unit of Population Epidemiology, Division of Primary Care Medicine, Geneva University Hospitals, Geneva, Switzerland), Chantal Martinez (Unit of Population Epidemiology, Division of Primary Care Medicine, Geneva University Hospitals, Geneva, Switzerland), Paola D'Ippolito (Unit of Population Epidemiology, Division of Primary Care Medicine, Geneva University Hospitals, Geneva, Switzerland), Camille Tible (Unit of Population Epidemiology, Division of Primary Care Medicine, Geneva University Hospitals, Geneva, Switzerland), Viola Bucolli (Unit of Population Epidemiology, Division of Primary Care Medicine, Geneva University Hospitals, Geneva, Switzerland), Livia Boehm (Unit of Population Epidemiology, Division of Primary Care Medicine, Geneva University Hospitals, Geneva, Switzerland), Adrien Jos Rastello (Unit of Population Epidemiology, Division of Primary Care Medicine, Geneva University Hospitals, Geneva, Switzerland), Lucie Ménard (Médecine & Hygiène, 1225 Chêne-Bourg), Lison Beigbeder (Médecine & Hygiène, 1225 Chêne-Bourg), Fréderic Rinaldi (dotBase, 1227 Geneva), Alain Cudet (dotBase, 1227 Geneva), Alexandre Moulin (dotBase, 1227 Geneva), Andrew S Azman (Unit of Population Epidemiology, Division of Primary Care Medicine, Geneva University Hospitals, Geneva, Switzerland; Institute of Global Health, Faculty of Medicine, University of Geneva, Geneva, Switzerland; Department of Epidemiology, Johns Hopkins Bloomberg School of Public Health, Baltimore, USA), Arnaud G L'Huillier (Laboratory of Virology, Department

of Diagnostics, Geneva University Hospitals, Geneva, Switzerland; Department of Pediatrics, Gynecology & Obstetrics, Pediatric Infectious Disease Unit, Geneva University Hospitals and Faculty of Medicine, Geneva, Switzerland), Klara M Posfay-Barbe (Department of Pediatrics, Gynecology & Obstetrics, Pediatric Infectious Disease Unit, Geneva University Hospitals and Faculty of Medicine, Geneva, Switzerland), Idris Guessous (Division of Primary Care, Geneva University Hospitals, Geneva, Switzerland; Department of Health and Community Medicine, Faculty of Medicine, University of Geneva, Geneva, Switzerland), Silvia Stringhini (Unit of Population Epidemiology, Division of Primary Care Medicine, Geneva University Hospitals, Geneva, Switzerland; Department of Health and Community Medicine, Faculty of Medicine, University of Geneva, Geneva, Switzerland; University Center for General Medicine and Public Health, University of Lausanne, Lausanne, Switzerland), Grégoire Michielin (Institute of Bioengineering, School of Engineering, École Polytechnique Fédérale de Lausanne, Lausanne, Switzerland), Sebastian J Maerkl (Institute of Bioengineering, School of Engineering, École Polytechnique Fédérale de Lausanne, Lausanne, Switzerland), Fatemeh Arefi (Institute of Bioengineering, School of Engineering, École Polytechnique Fédérale de Lausanne, Lausanne, Switzerland), Mathilde Bellon (Department of Microbiology and Molecular Medicine, Faculty of Medicine, University of Geneva, Geneva, Switzerland; Center for Emerging Viral Diseases, Geneva University Hospitals and University of Geneva, Geneva, Switzerland), Isabella Eckerle (Department of Microbiology and Molecular Medicine, Faculty of Medicine, University of Geneva, Geneva, Switzerland; Center for Emerging Viral Diseases, Geneva University Hospitals and University of Geneva, Geneva, Switzerland; Laboratory of Virology, Department of Diagnostics, Geneva University Hospitals, 1205 Geneva, Switzerland; Division of Infectious Diseases, Department of Medicine, Geneva University Hospitals, 1205 Geneva, Switzerland), Laurent Kaiser (Center for Emerging Viral Diseases, Geneva University Hospitals and University of Geneva, Geneva, Switzerland; Laboratory of Virology, Department of Diagnostics, Geneva University Hospitals, 1205 Geneva, Switzerland; Division of Infectious Diseases, Department of Medicine, Geneva University Hospitals, 1205 Geneva, Switzerland), Benjamin Meyer (Centre for Vaccinology, Department of Pathology and Immunology, University of Geneva, Geneva, Switzerland), Meriem Bekliz (Department of Microbiology and Molecular Medicine, Faculty of Medicine, University of Geneva, Geneva, Switzerland; Center for Emerging Viral Diseases, Geneva University Hospitals and University of Geneva, Geneva, Switzerland), Florian Laubscher (Laboratory of Virology, Department of Diagnostics, Geneva University Hospitals, 1205 Geneva, Switzerland), Francisco Perez Rodriguez (Center for Emerging Viral Diseases, Geneva University Hospitals and University of Geneva, Geneva, Switzerland), Pascale Sattonnet-Roche (Department of Microbiology and Molecular Medicine, Faculty of Medicine, University of Geneva, Geneva, Switzerland; Center for Emerging Viral Diseases, Geneva University Hospitals and University of Geneva, Geneva, Switzerland), Catia Alvarez (Department of Microbiology and Molecular Medicine, Faculty of Medicine, University of Geneva, Geneva, Switzerland; Center for Emerging Viral Diseases, Geneva University Hospitals and University of Geneva, Geneva, Switzerland), Kenneth Adea (Department of Microbiology and Molecular Medicine, Faculty of Medicine, University of Geneva, Geneva, Switzerland; Center for Emerging Viral Diseases, Geneva University Hospitals and University of Geneva, Geneva, Switzerland), Manel Essaidi-Laziosi (Department of Microbiology and Molecular Medicine, Faculty of Medicine, University of Geneva, Geneva, Switzerland; Center for Emerging Viral Diseases, Geneva University Hospitals and University of Geneva, Geneva, Switzerland), Gil Barbosa Monteiro (Laboratory of Virology, Department of Diagnostics, Geneva University Hospitals, 1205 Geneva, Switzerland)

## Author Contributions

**Conceptualization:** Andrew S. Azman, Arnaud G. L'Huillier, Klara M. Posfay-Barbe, Laurent Kaiser, Idris Guessous, Sebastian J. Maerkl, Isabella Eckerle, Silvia Stringhini.

**Data curation:** Elsa Lorthe, Francesco Pennacchio, Florian Laubscher, Fatemeh Arefi.

**Formal analysis:** Elsa Lorthe, Grégoire Michielin, Meriem Bekliz, Florian Laubscher, Fatemeh Arefi, Javier Perez-Saez.

**Funding acquisition:** Sebastian J. Maerkl, Isabella Eckerle, Silvia Stringhini.

**Investigation:** Elsa Lorthe, Mathilde Bellon, Julie Berthelot, Silvia Stringhini.

**Methodology:** Elsa Lorthe, Grégoire Michielin, María-Eugenia Zaballa, Francesco Pennacchio, Meriem Bekliz, Andrew S. Azman, Arnaud G. L'Huillier, Klara M. Posfay-Barbe, Laurent Kaiser, Idris Guessous, Sebastian J. Maerkl, Isabella Eckerle, Silvia Stringhini.

**Project administration:** Elsa Lorthe, Julie Berthelot, María-Eugenia Zaballa, Silvia Stringhini.

**Resources:** Silvia Stringhini.

**Supervision:** Silvia Stringhini.

**Validation:** Mathilde Bellon, Isabella Eckerle, Silvia Stringhini.

**Visualization:** Javier Perez-Saez.

**Writing – original draft:** Elsa Lorthe.

**Writing – review & editing:** Mathilde Bellon, Grégoire Michielin, Julie Berthelot, María-Eugenia Zaballa, Francesco Pennacchio, Meriem Bekliz, Florian Laubscher, Fatemeh Arefi, Javier Perez-Saez, Andrew S. Azman, Arnaud G. L'Huillier, Klara M. Posfay-Barbe, Laurent Kaiser, Idris Guessous, Sebastian J. Maerkl, Isabella Eckerle, Silvia Stringhini.

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
