## [Decision Letter · Decision Letter 0]

3 May 2022

PONE-D-22-07591Epidemiological, virological and serological investigation of a SARS-CoV-2 outbreak (Alpha variant) in a primary school: a prospective longitudinal studyPLOS ONE

Dear Dr. Lorthe,

Thank you for submitting your manuscript to PLOS ONE. After careful consideration, we feel that it has merit but does not fully meet PLOS ONE’s publication criteria as it currently stands. Therefore, we invite you to submit a revised version of the manuscript that addresses the points raised during the review process. 

We look forward to receiving your revised manuscript.

Kind regards,

Binod Kumar, PhD

Academic Editor

PLOS ONE

Journal Requirements:

3. One of the noted authors is a group or consortium the SEROCoV-Schools Study Group. In addition to naming the author group, please list the individual authors and affiliations within this group in the acknowledgments section of your manuscript. Please also indicate clearly a lead author for this group along with a contact email address.

4. Please ensure that you refer to Figure 4 in your text as, if accepted, production will need this reference to link the reader to the figure.

Reviewers' comments:

Reviewer's Responses to Questions

**Comments to the Author**

1. Is the manuscript technically sound, and do the data support the conclusions?

Reviewer #1: Yes

Reviewer #2: Yes

2. Has the statistical analysis been performed appropriately and rigorously? 

Reviewer #1: Yes

Reviewer #2: Yes

3. Have the authors made all data underlying the findings in their manuscript fully available?

Reviewer #1: Yes

Reviewer #2: Yes

4. Is the manuscript presented in an intelligible fashion and written in standard English?

Reviewer #1: Yes

Reviewer #2: Yes

5. Review Comments to the Author

Reviewer #1: In the present prospective study titled “Epidemiological, virological and serological investigation of a SARS-CoV-2 outbreak (Alpha variant) in a primary school: a prospective longitudinal study” the author tried to address the transmission of SARS-CoV-2 from school to households. This study investigated SARS-CoV-2 outbreak in a primary school in Geneva, Switzerland in the months of April and May in the year 2021. The study is well designed and describes the transmission dynamics of SARS-CoV-2 from school children to households. They used well established diagnostic protocols such as rapid antigen test, RT-PCR, serological anti-spike antibodies and Viral genome sequencing for the identification and tracing infection.

The only limitation to this study is the sample size which the author also agreed in their discussion. However, this kind of study’s are important in implementing effective strategies to limit the transmission.

Reviewer #2: This manuscript upholds children and schools as a potential strong contributor of the overall community SARS-CoV-2 transmission which is highly relevant as many Lockdown protocols suspended schools or switched to remote learning.

Minor comments:

The results and conclusions are too short. There is no information on peak infectivity period of open cases. How long the student/teacher were at school before symptoms appear i.e. was there asymptomatic spread and if any signs or symptoms were subjects quarantined or home isolated (did subjects attend school while waiting for the tests). During contact tracing was it all suspected contact transmission, droplets or aerosol transmission? Authors do mention “quarantine by local health authorities for 10 days” does that apply to the whole class together or different individuals according to SARS-CoV-2 positivity/visible clinical symptoms?

Authors need to describe abbreviations whenever used for the first time including obvious D0-1.

Major comments:

The term “verbal consent” by children is confusing as children do not have the legal capacity to “consent” to participate in research. The proper terminology is “Ascent” which just needs paraphrasing as it is what researchers did. Authors may want to mention how the children were informed e.g. videos, graphics, and other visual aids to help explain the trial along with that fact that it is a minimal risk category trial.

Decision:

This is a very systematic and well laid out study. It’s unusual for going with fixed testing scheme for COVID-19 testing trials. Authors have explained the settings and limitations very well and leaves readers to compare their study findings. With so many studies published on alpha virus dynamics, the publication does seem dissociated as there is scarce literature review in both introduction as well as discussion sections.

With much more high transmission SARS-CoV-2 strains circulating and the rapid changes in the dynamics, public health is in continuous pressure and risky updates. It would be prudent if some take home message is added in context of current scenario from findings of this study.

6. PLOS authors have the option to publish the peer review history of their article (what does this mean?). If published, this will include your full peer review and any attached files.

Reviewer #1: **Yes: **Ella Bhagyaraj

Reviewer #2: **Yes: **Yash Gupta

---

## [Author Response · Author response to Decision Letter 0]

7 Jun 2022

Journal Requirements: When submitting your revision, we need you to address these additional requirements.

We have adapted our manuscript to meet PLOS ONE’s style requirements.

Despite all the precautions taken, our data contain potentially identifying information due to the nature of this study and the fact that we are reporting information on relatively small groups of children from a specific geographic location. Therefore, the study steering committee members decided to make these data accessible to researchers who meet the criteria for access to confidential data upon reasonable request for data sharing to the Unit of Population Epidemiology (uep@hcuge.ch). All requests will be evaluated by the Data Access Committee and approved on the basis of their scientific quality.

3. One of the noted authors is a group or consortium the SEROCoV-Schools Study Group. In addition to naming the author group, please list the individual authors and affiliations within this group in the acknowledgments section of your manuscript. Please also indicate clearly a lead author for this group along with a contact email address.

The lead author of the SEROCoV-Schools group has been mentioned in the group list, with an email address. All individual affiliations have been added in the acknowledgements section, as required.

4. Please ensure that you refer to Figure 4 in your text as, if accepted, production will need this reference to link the reader to the figure.

The reference to Figure 4 has been added in the manuscript.

We have checked the references list, and have updated a reference from a pre-print that is now published. We have also added references in response to the reviewer’s advice to provide a more extensive literature review.

Comments to the Author

Reviewer #1: In the present prospective study titled “Epidemiological, virological and serological investigation of a SARS-CoV-2 outbreak (Alpha variant) in a primary school: a prospective longitudinal study” the author tried to address the transmission of SARS-CoV-2 from school to households. This study investigated SARS-CoV-2 outbreak in a primary school in Geneva, Switzerland in the months of April and May in the year 2021. The study is well designed and describes the transmission dynamics of SARS-CoV-2 from school children to households. They used well established diagnostic protocols such as rapid antigen test, RT-PCR, serological anti-spike antibodies and Viral genome sequencing for the identification and tracing infection. The only limitation to this study is the sample size which the author also agreed in their discussion. However, this kind of study’s are important in implementing effective strategies to limit the transmission.

We would like to thank the reviewer for this positive evaluation of our work. We acknowledge a limited sample size, however, by definition, the sample size of a school class outbreak can only be as large as the number of pupils in each school class. As participation was high, we were in the best possible conditions to study this outbreak.

Reviewer #2: This manuscript upholds children and schools as a potential strong contributor of the overall community SARS-CoV-2 transmission which is highly relevant as many Lockdown protocols suspended schools or switched to remote learning.

Minor comments:

The results and conclusions are too short. 

We consider these two sections to be quite comprehensive. We welcome any suggestions for topics that should be covered further.

There is no information on peak infectivity period of open cases. How long the student/teacher were at school before symptoms appear i.e. was there asymptomatic spread and if any signs or symptoms were subjects quarantined or home isolated (did subjects attend school while waiting for the tests).

Unfortunately, the design of this study did not allow to estimate the peak infectivity period: once a participant tested positive, he/she was not retested at subsequent visits of the same infection. Peak infectivity, however, has been described in previous studies and seems to occur on the day of symptom onset or a few days before (https://doi.org/10.1038/s41591-020-0869-5). As mentioned in the article, 2 children were absent from school for 2 weeks prior to this outbreak and were not included in the investigation, while 2 symptomatic children were absent from school when we organized the first tests, both were tested by healthcare providers outside the study setting and had a positive RT-PCR. All other participants were attending school as usual. The first teacher who tested positive was at school on the Friday when symptoms started, but then stayed home until the positive test and 10 days afterwards. We added this information in the manuscript: “Both attended school until April 23 included.” and “They all attended school the week before.”.

As there were no specific recommendations from the health authorities at the outbreak onset, children attended school as usual before the tests (that we were able to organize very quickly after the first case was notified). Importantly, the results of the antigen rapid diagnostic tests performed by our team were available in 15 minutes, and were communicated immediately to the families. The results of the RT-PCR tests were communicated within 24 hours. Of course, all subjects who tested positive were home isolated for 10 days. Information on the communication of results was added in the manuscript: “All results were communicated to the participants as soon as they became available.”, “Serologies were communicated to the families approximately 2 months after the visit.”.

Therefore, asymptomatic spread likely happened.

During contact tracing was it all suspected contact transmission, droplets or aerosol transmission?

We did not aim to determine the mode of transmission (contact, droplets or aerosol) with this study. As a consequence, we did not focus our investigation and contact tracing interviews on this point. All we can say is that close contacts occurred both at school and at home, and that young children were not wearing face masks. This last information was added in the manuscript: “In the investigated school, measures in place during the outbreak included checking children’s temperature every morning, sending children home if they had fever or sickness beyond very mild symptoms, restricted access for parents and requiring all adults (but not young children) to wear facemasks.”

Authors do mention “quarantine by local health authorities for 10 days” does that apply to the whole class together or different individuals according to SARS-CoV-2 positivity/visible clinical symptoms?

The local health authorities decided to quarantine all children in these two classes for 10 days. We have clarified this point in the manuscript: “Of note, all children from these two classes were placed in quarantine by local health authorities for 10 days from April 30, 2021.”

Authors need to describe abbreviations whenever used for the first time including obvious D0-1.

We have removed all abbreviations related to Days for ease of reading.

Major comments:

The term “verbal consent” by children is confusing as children do not have the legal capacity to “consent” to participate in research. The proper terminology is “Ascent” which just needs paraphrasing as it is what researchers did. 

Following the reviewer’s advice, we have replaced consent with assent. 

Authors may want to mention how the children were informed e.g. videos, graphics, and other visual aids to help explain the trial along with that fact that it is a minimal risk category trial. 

Teachers, school staff and parents were informed both orally and through a set of documents including a flyer and a detailed information note. We also provided a short letter written in a child-friendly style, and a schema illustrating the concepts of infection and antibodies, which parents could use to inform their child. Before each test, a team of trained nurses discussed with each child, explained the tests to be taken and answered any question children may have had. The test was then performed only if the child agreed it. Because most participants were very young children, we chose minimally invasive tests.

We have detailed these elements in the manuscript: 

“All parents and teachers were informed about the study (orally and through a flyer and a detailed information note) and gave written consent. We also provided a short letter written in a child-friendly style, and a schema illustrating concepts of infection and antibodies, which parents could use to inform their child. Before each test, a team of trained nurses discussed with each child, explained the tests to be taken and answered any questions. Children gave verbal consent assent to participate.”

 “This swabbing method is less invasive than nasopharyngeal swabs and more acceptable by young children.”

“This sampling method is minimally invasive, easily implementable in schools and more acceptable than traditional venipuncture by children.”

Decision:

This is a very systematic and well laid out study. It’s unusual for going with fixed testing scheme for COVID-19 testing trials. Authors have explained the settings and limitations very well and leaves readers to compare their study findings. With so many studies published on alpha virus dynamics, the publication does seem dissociated as there is scarce literature review in both introduction as well as discussion sections.

To respond to the concern raised by the reviewer, we have revised the introduction and discussion sections and have added relevant references:

“Published studies reported school outbreaks in the United States, Australia, England, Ireland, Norway or Italy, and/or seroprevalence estimates in schools in Switzerland, Canada or Germany. Overall, they suggested a relatively low incidence of COVID-19 in children or low prevalence of anti-SARS-CoV-2 antibodies, conveying the message that schools are not a high risk setting for transmission of COVID-19.”

“Similar patterns were found in a prospective study of three B.1.1.7 variant outbreaks which occurred in three childcare centers in Germany in January–February 2021: 11/12 groups had secondary cases, and secondary attack rates ranged from 7 to 64% among groups of children aged 1-6 years.”

“This finding is consistent with prospective research based on comprehensive testing protocols which identified a higher incidence of COVID-19 among children and higher rates of transmission than previously reported, including in households.”

With much more high transmission SARS-CoV-2 strains circulating and the rapid changes in the dynamics, public health is in continuous pressure and risky updates. It would be prudent if some take home message is added in context of current scenario from findings of this study.

Thank you for this relevant suggestion. We have added the following sentence at the end of the discussion: “As a take-home message, we suggest that routinely reported infections are likely to be just the tip of the iceberg, highlighting the importance of testing contacts, regardless of symptoms, to identify SARS-CoV-2 infections and stop transmission by rapidly isolating positive cases.”

---

## [Decision Letter · Decision Letter 1]

7 Jul 2022

PONE-D-22-07591R1Epidemiological, virological and serological investigation of a SARS-CoV-2 outbreak (Alpha variant) in a primary school: a prospective longitudinal studyPLOS ONE

Dear Dr. Lorthe,

Thank you for submitting your manuscript to PLOS ONE. After careful consideration, we feel that it has merit but does not fully meet PLOS ONE’s publication criteria as it currently stands. Therefore, we invite you to submit a revised version of the manuscript that addresses the points raised during the review process.

ACADEMIC EDITOR:Please revise the paper as per the minor recommendations by the reviewer.

We look forward to receiving your revised manuscript.

Kind regards,

Debdutta Bhattacharya

Academic Editor

PLOS ONE

Journal Requirements:

Reviewers' comments:

Reviewer's Responses to Questions

**Comments to the Author**

1. If the authors have adequately addressed your comments raised in a previous round of review and you feel that this manuscript is now acceptable for publication, you may indicate that here to bypass the “Comments to the Author” section, enter your conflict of interest statement in the “Confidential to Editor” section, and submit your "Accept" recommendation.

Reviewer #1: All comments have been addressed

Reviewer #2: All comments have been addressed

2. Is the manuscript technically sound, and do the data support the conclusions?

Reviewer #1: Partly

Reviewer #2: Yes

3. Has the statistical analysis been performed appropriately and rigorously? 

Reviewer #1: Yes

Reviewer #2: Yes

4. Have the authors made all data underlying the findings in their manuscript fully available?

Reviewer #1: (No Response)

Reviewer #2: Yes

5. Is the manuscript presented in an intelligible fashion and written in standard English?

Reviewer #1: Yes

Reviewer #2: Yes

6. Review Comments to the Author

Reviewer #1: The author addressed my concerns and improved the manuscript based on the reviewers comments and incorporated the changes.

Reviewer #2: All comments have been reasonably and meticulously addressed.

The limitations of the study have been clearly updated.

7. PLOS authors have the option to publish the peer review history of their article (what does this mean?). If published, this will include your full peer review and any attached files.

Reviewer #1: No

Reviewer #2: **Yes: **Yash Gupta

---

## [Author Response · Author response to Decision Letter 1]

22 Jul 2022

This is a re-submission of my R1 revision. I received an email on July 7 asking for additional revisions, but the reviewers were satisfied with R1 and did not recommend any further revisions, so none have been made. There was apparently a technical error when the academic editor submitted their decision.

---

## [Editor Report · Decision Letter 2]

25 Jul 2022

Epidemiological, virological and serological investigation of a SARS-CoV-2 outbreak (Alpha variant) in a primary school: a prospective longitudinal study

PONE-D-22-07591R2

Dear Dr. Lorthe,

We’re pleased to inform you that your manuscript has been judged scientifically suitable for publication and will be formally accepted for publication once it meets all outstanding technical requirements.

Kind regards,

Debdutta Bhattacharya

Academic Editor

PLOS ONE
---

## [Editor Report · Acceptance letter]

8 Aug 2022

PONE-D-22-07591R2 

Epidemiological, virological and serological investigation of a SARS-CoV-2 outbreak (Alpha variant) in a primary school: a prospective longitudinal study 

Dear Dr. Lorthe:

I'm pleased to inform you that your manuscript has been deemed suitable for publication in PLOS ONE. Congratulations! Your manuscript is now with our production department. 

Kind regards, 

on behalf of

Dr. Debdutta Bhattacharya 

Academic Editor

PLOS ONE